# Transcriptome Profiling Identifies Candidate Genes Contributing to Male and Female Gamete Development in Synthetic *Brassica* Allohexaploids

**DOI:** 10.3390/plants11121556

**Published:** 2022-06-13

**Authors:** Chengyan Ji, Zhaoran Tian, Yue Liu, Gongyao Shi, Baoming Tian, Weiwei Chen, Zhengqing Xie, Xingzhou Han, Niannian Liang, Fang Wei, Xiaochun Wei

**Affiliations:** 1Henan International Joint Laboratory of Crop Gene Resources and Improvements, School of Agricultural Sciences, Zhengzhou University, Zhengzhou 450001, China; jichengyan98@163.com (C.J.); tzr1369@126.com (Z.T.); ly1274087232@163.com (Y.L.); shigy@zzu.edu.cn (G.S.); tianbm@zzu.edu.cn (B.T.); weiwei_chen15134@zzu.edu.cn (W.C.); zqxie@zzu.edu.cn (Z.X.); hanxingzhou2021@163.com (X.H.); lw18336262927@163.com (N.L.); 2Institute of Horticulture, Henan Academy of Agricultural Sciences, Graduate T & R Base of Zhengzhou University, Zhengzhou 450002, China

**Keywords:** transcriptome, *Brassica*, allohexaploid, polyploid, male and female gamete development, meiosis, anther, ovule

## Abstract

Polyploidy plays a crucial role in plant evolution and speciation. The development of male and female gametes is essential to the reproductive capacity of polyploids, but their gene expression pattern has not been fully explored in newly established polyploids. The present study aimed to reveal a detailed atlas of gene expression for gamete development in newly synthetic *Brassica* allohexaploids that are not naturally existing species. Comparative transcriptome profiling between developing anthers (staged from meiosis to mature pollen) and ovules (staged from meiosis to mature embryo sac) was performed using RNA-Seq analysis. A total of 8676, 9775 and 4553 upregulated differentially expressed genes (DEGs) were identified for the development of both gametes, for male-only, and for female-only gamete development, respectively, in the synthetic *Brassica* allohexaploids. By combining gene ontology (GO) biological process analysis and data from the published literature, we identified 37 candidate genes for DNA double-strand break formation, synapsis and the crossover of homologous recombination during male and female meiosis and 51 candidate genes for tapetum development, sporopollenin biosynthesis and pollen wall development in male gamete development. Furthermore, 23 candidate genes for mitotic progression, nuclear positioning and cell specification and development were enriched in female gamete development. This study lays a good foundation for revealing the molecular regulation of genes related to male and female gamete development in *Brassica* allohexaploids and provides more resourceful genetic information on the reproductive biology of *Brassica* polyploid breeding.

## 1. Introduction

Polyploidy has long been recognized as an important evolutionary force in plants [1,2,3]. The formation of polyploids is attributed to genomic plasticity, and polyploid-induced changes can result in new genetic diversity and advantageous adaptations to the environment [4,5]. Polyploids generally have multiple phenotypes and greater growth vigor compared with their parents [6]. Polyploidy also leads to an increase in plant organs to improve outputs and adapt to various biological and abiotic stresses in plant breeding [7,8]. Allopolyploidy results concomitantly from the genome double after the hybridization of two or more species [9]. *Brassica* has long been considered as a model to explore polyploidization. Three basic diploid species in *Brassica*, *Brassica rapa* (AA, 2n = 20), *Brassica nigra* (BB, 2n = 16) and *Brassica oleracea* (CC, 2n = 18), have hybridized to give rise to three allotetraploid species, *Brassica juncea* (AABB, 2n = 36), *Brassica napus* (AACC, 2n = 38) and *Brassica carinata* (BBCC, 2n = 34). The combination of genetic variation from six species in *Brassica* could result in crops with increased adaptation and agronomic potential as well as improved heterosis from the contribution of alleles [10,11,12]. *Brassica* allohexaploids (2n = AABBCC) do not exist in nature but can be synthesized by hybridization between diploid and/or allotetraploid species of *Brassica*. The cross between *B. carinata* and *B. rapa* is the most commonly attempted and successful method in the five possible species combinations that can produce *Brassica* allohexaploids [13]. Hybridization and genomic doubling may lead to extensive transcriptomic changes in the synthesized trigenomic *Brassica* allohexaploids relative to their parents [14]. *Brassica* allohexaploids not only provide an effective way to improve the genetic diversity of *Brassica* and the intergenic hybridization of oilseed crops in the future, but they also offer excellent material for the study of polyploid plants.

So far, studies on *Brassica* allohexaploids have mainly focused on agronomic traits, meiotic behaviors and subgenome stabilities [15,16]. The development of male and female gametes is crucial to the reproductive capacity of sexually reproducing organisms, especially polyploids, and is strictly regulated by complex processes. However, the genes involved in male and female gamete development are not fully understood in *Brassica* allohexaploids. Therefore, it is very important to establish a reproductive transcriptome profiling of *Brassica* allohexaploids.

The male germline matures within the anther, whereas the female germline develops within the ovule [17]. The male and female gametophyte are the important reproductive units of angiosperms and are essential for sexual reproduction [18,19]. Both male and female germline development of angiosperms consist of two main stages: microsporogenesis and microgametogenesis, giving rise to male gametes, and megasporogenesis and megagametogenesis, leading to the formation of female gametes. Microsporogenesis and megasporogenesis are key to male and female reproduction and require the completion of meiosis to form microspores and functional megaspores (FM). Microgametogenesis includes an asymmetric division to form vegetative and generative cells, and the generative cells produce two male gametes after a mitotic division [20]. FM undergoes three rounds of mitosis, nuclear migration, cellularization and differentiation to form a mature seven-celled embryo sac, containing three antipodal cells, two synergid cells, one egg cell and one central cell that contains two polar nuclei, which complete megagametogenesis [19,21]. During gamete generation, male and female gametes have similar ploidy transition and cell cycle progressions, while they show many differences in gamete development and specialization. 

RNA-Seq technology is used to analyze the structure and function of genes at the organismal level and to explore a range of biological pathways [22,23]. The establishment of *Brassica* transcriptome databases, such as the *B. napus* transcriptome database BnTIR, provide a lot of useful resources for the study of *Brassica* polyploidy at the transcriptional level [24]. In recent years, RNA-Seq has been successfully used in anther and ovule development in many species. In *Brassica napus* L., lipid metabolism genes involved in pollen extine formation, elaioplast and tapetosome biosynthesis were preferentially expressed in early anthers, and carbohydrate metabolism genes to form pollen intine and to accumulate starch in mature pollen grains were preferentially expressed in late anthers [25]. Comparative analysis of differential gene expression revealed multiple signal pathways during flowering of autotetraploid *B. rapa* [26]. The molecular processes involved in the development of female gametes in plants are much less understood than those involved in the development of male gametes due to the small number and difficult availability of female gametophytes. Using high-throughput sequencing analysis, female gamete development in *Arabidopsis* has been intensively studied, and a number of key genes regulating megasporogenesis and megagametogenesis have been found, offering fundamental knowledge of these developmental processes [21,27]. 

The aim of the study was to reveal a detailed atlas of gene expression for gamete development in synthetic *Brassica* allohexaploids. We performed differential expression analysis to identify common and preferential genes that regulate the developmental events of male and female gametes. This study provides rich genetic resources for the cloning and functional verification of genes related to male and female gamete development and lays a good foundation for revealing the molecular regulatory mechanism of reproductive development in *Brassica* allohexaploids.

## 2. Results

### 2.1. Transcriptome Sequencing and Sequence Alignment

RNA-Seq of anthers from meiosis to the mature pollen stage (Anther), ovules from meiosis to mature embryo sac stage (Ovule) and young leaves (Leaf) was performed by Illumina in synthetic *Brassica* allohexaploids. Leaf was used as vegetative tissue (organ) control, and each *Brassica* allohexaploid tissue (organ) received three biological replicates. RNA-Seq results are presented in Table 1 and Table 2. The number of clean reads from the nine RNA-Seq libraries ranged from 40,202,856 to 47,450,126 (Table 1). The Clean Q30 Bases Rate was greater than 94.29%, and the higher the value is, the better the sequencing quality (Table 1). All clean reads were then aligned to *B. rapa*, *B. nigra* and *B. oleracea* genome sequences using HISAT2 software. Mapped genome reads ranged from 21,502,559 to 39,794,184, and genome mapping rates ranged from 53.49% to 85.19% (Table 2). These results suggested that their quality met the requirements for transcriptome sequencing. Fragments per kilobase per million reads (FPKM) were determined for all genes of Anther, Ovule and Leaf in *Brassica* allohexaploids (Appendix A). Correlation heat maps of the transcriptome from Anther, Ovule and Leaf showed that the three biological replicates of each tissue were grouped together with high correlation (Appendix A). These results indicated that the three biological replicates of per tissue had good repeatability in this study.

### 2.2. Anthers at Male Gamete Development Stage and Ovules at Female Gamete Development Stage Contained More Genes Than Leaves in Brassica Allohexaploids

A total of 100,804 genes were identified in the RNA-Seq data of *Brassica* allohexaploids, among which 96,286, 86,712 and 79,875 genes were expressed in Anther, Ovule and Leaf, respectively, providing sufficient data for studying male and female gamete development (Figure 1a). Among the 100,804 genes, 74,597 genes were expressed in all three tissues, and 9948, 1870 and 1514 genes were specifically expressed in Anther, Ovule and Leaf, respectively (Figure 1a). Comparative analysis showed that Anther and Ovule overlapped most, with 9111 genes commonly expressed in these two tissues; 2630 genes were commonly expressed in Anther and Leaf; and 1134 genes were commonly expressed in Ovule and Leaf (Figure 1a). Anther and Ovule contained more genes and more specific genes than vegetative tissue Leaf. In all three tissues, transcript abundance analysis indicated 74.1% to 78.6% genes with low expression (FPKM >0 to ≤10) and 19.6% to 24.0% genes with moderate expression (FPKM >10 to ≤100) but only 1.4% to 1.9% genes with high expression (FPKM > 100) (Figure 1b). 

In Anther, Ovule and Leaf, 33,837, 30,501 and 28,078 genes, respectively, were expressed from A-genome; 35,126, 31,413 and 28,842 genes, respectively, were expressed from B-genome; and 27,323, 24,798 and 22,955 genes, respectively, were expressed from C-genome (Figure 1c). The reference genomes of *B. rapa*, *B. nigra* and *B. oleracea* contained 41,020, 47,953 and 61,279 genes, respectively, thus the percentage of genes expressed in Anther, Ovule and Leaf from A-genome was 82.5%, 74.4% and 68.4%, respectively, while from B-genome the values were 73.3%, 65.5% and 60.1%, respectively, and from C-genome 44.6%, 40.5% and 37.5% (Figure 1c), respectively. These results indicated that the B-genome contained the largest number of genes in Anther, Ovule and Leaf in *Brassica* allohexaploids, while the A-genome was bias expressed. Three groups of differentially expressed genes (DEGs) were generated by pairing differential expression analysis of genes between all three tissues (Appendix A). Volcano plots show DEGs of Anther vs. Ovule, Ovule vs. Leaf and Anther vs. Leaf (Appendix A). Among all comparisons, the number of DEGs in Anther and Ovule was the smallest, and 22,305 genes (14,366 upregulated and 7939 downregulated) were differentially expressed in Anther compared with Ovule (Figure 1d). Compared with Leaf, 30,893 genes (14,830 upregulated and 16,063 downregulated) were differentially expressed in Ovule (Figure 1d). However, 35,849 genes (19,513 upregulated and 16,336 downregulated) were differentially expressed in Anther compared with Leaf, which was the largest number of DEGs (Figure 1d). The findings showed that there were differences in gene expression between reproductive tissue and vegetative tissue and male gamete development had more genes expressed than female gamete development in *Brassica* allohexaploids. 

### 2.3. Identification of Upregulated Genes Co-Expressed and Preferentially Expressed Genes in Male and Female Gamete Development of Brassica Allohexaploids

In order to study the common characteristics of male and female gamete development in *Brassica* allohexaploids, the 19,513 upregulated genes in Anther relative to Leaf and the 14,830 upregulated genes in Ovule relative to Leaf were overlapped. There were 25,667 upregulated genes co-expressed and 8676 preferentially expressed genes in male and female gamete development of *Brassica* allohexaploids (Figure 2a); 8605 of the 25,667 upregulated genes co-expressed in male and female gamete development were annotated using clusters of orthologous groups (COG) classification and functionally divided into 25 COG categories (Figure 2b). COG clusters were mainly showed in General function prediction only [1494~13.43%]; Transcription [1079~9.70%]; Posttranslational modification, protein turnover, chaperones [861~7.74%]; Replication, recombination and repair [778~6.99%]; Carbohydrate transport and metabolism [775~6.97%]; Signal transduction mechanisms [590~5.30%]; Translation, ribosomal structure and biogenesis [576~5.18%]; Cell cycle control, cell division, chromosome partitioning [559~5.03%]; Amino acid transport and metabolism [557~5.01%]; and Lipid transport and metabolism [470~4.23%] (Figure 2b). To identify the key biological process pathways of common genes in male and female gamete development of *Brassica* allohexaploids, 8676 preferentially expressed genes at male and female gamete development stages were analyzed by gene ontology (GO) biological process enrichment analysis. The significantly enriched GO terms had DNA repair (GO: 0006281), DNA replication (GO: 0006260), synapsis (GO: 0007129), cell cycle (GO: 0007049), microtubule-based movement (GO: 0007018), DNA recombination (GO: 0006310), double-strand break repair via homologous recombination (GO: 0000724), mismatch repair (GO: 0006298), resolution of meiotic recombination intermediates (GO: 0000712), double-strand break repair (GO: 0006302), chiasma assembly (GO: 0051026), double-strand break repair via synthesis-dependent strand annealing (GO: 0045003), homologous recombination (GO: 0035825), regulation of double-strand break repair (GO: 2000779) and regulation of double-strand break repair via homologous recombination (GO: 0010569) (Figure 2c). These results suggested that common genes in male and female gamete development may be related to the homologous recombination of meiosis in *Brassica* allohexaploids.

*Arabidopsis* orthologs of seven genes, BniB034973 (orthologous to *SPO11-1* (Sporulation 11-1)), BniB027699 and Bol025700 (orthologous to *PRD1* (PUTATIVE RECOMBINATION INITIATION DEFECT 1)), Bra024921, Bra033241 and Bol007924 (orthologous to *PRD3* (PUTATIVE RECOMBINATION INITIATION DEFECT 3)) and BniB044390 (orthologous to *MTOPVIB* (Meiotic topoisomerase VIB-like)) are necessary for meiotic double-strand breaks (DSBs) formation (Figure 3). *SPO11-1* is essential for formation of DSBs in plants, and *PRD1*, *PRD3* and *MTOPVIB* have already been shown to play a key role in DNA DSBs formation in *Arabidopsis* (Figure 3) [28,29,30,31]. In addition, some meiosis genes related to synapses are also enriched in the GO biological process during the development of male and female gametes. Bra004222 and BniB009412 are orthologous to *ASY1* (Asynaptic 1), which encodes a protein essential for meiotic chromosomal synapsis and localizes to axis-associated chromatin [32]. In *B. rapa*, the axis-associated mutant plants of Bra004222 produce fewer crossovers (COs) due to abnormalities in meiosis (Figure 3) [33]. BniB041399 is a homolog of *ASY3* (Asynaptic 3) (Figure 3). *AYS3* and *ASY1* have similar roles in *Arabidopsis*, and its deletion also disrupts synaptic complex (SC) formation [34]. Similarly, Bra003654, Bra035016 and Bol027500 are orthologs of *ZYP1a* encoding synaptonemal complex protein, which regulates chromosome synapsis and normal fidelity of crossing over (Figure 3) [35]. The orthologs of these candidate genes, Bra039674, Bra038126, BniB027427, BniB026849, BniB009174, Bol039109 and Bol035122 (orthologous to *HEI10* (Homolog of human hei10)), Bra029127 (orthologous to *SHOC1* (Shortage in chiasmata 1)) and Bra037488, BniB000031 and Bol043105 (orthologous to *SPO22*) are ZMM proteins, which are required for the formation of class I meiotic COs (Figure 3) [36,37,38]. The *Arabidopsis* orthologous gene *MSH5* (Muts homolog 5) of Bra035777 and BniB043474 partners orthologous gene *MSH4* (Muts homolog 4) of Bol032857 in class I meiotic CO pathway [39]. The *Arabidopsis* ortholog *TOP3a* (Topoisomerase 3alpha) of BniB047025 and *RMI1* (RecQ mediated instability 1) of Bra035838, Bra031944 and BniB045608 maintain genome stability by limiting CO formation in favor of non-crossover (NCO) events (Figure 3) [40,41]. Bra034416, BniB042826 and Bol031970 are orthologous to *FANCM* (Fanconi anemia complementation group M), a highly conserved helicase, which functions as a major factor limiting meiotic CO formation (Figure 3) [42]. *RECQ4A*, an *Arabidopsis* ortholog of Bra018418, Bra031741 and Bol022020, has been found to play a key role in DNA repair and homologous recombination suppression (Figure 3) [43]. 

### 2.4. Identification of Preferentially Expressed Genes in Male Gamete Development of Brassica Allohexaploids

To better understand the development of male gametes in *Brassica* allohexaploids, we analyzed the preferentially expressed genes in male gamete development. Venn diagrams showed that the overlap of 14,366 upregulated genes in Anther vs. Ovule and 19,513 upregulated genes in Anther vs. Leaf was 9775 preferentially expressed genes in male gamete development (Figure 4a). We used GO annotation to functionally classify these 9775 genes based on their biological process. We found that GO terms associated with the activation of protein kinase activity (GO: 0032147), protein ubiquitination (GO: 0016567), pollen wall assembly (GO: 0010208), anther wall tapetum development (GO: 0048658), pollen exine formation (GO: 0010584), sporopollenin biosynthetic process (GO: 0080110), pollen development (GO: 0009555), callose deposition in cell walls (GO: 0052543), anther development (GO: 0048653) and pollen sperm cell differentiation (GO: 0048235) were significantly overrepresented within the preferentially expressed genes in male gamete development (Figure 4b). These findings showed that preferentially expressed genes in male gamete development were concentrated in pollen wall formation and other related pathways in *Brassica* allohexaploids.

The transcription factor *AMS* (ABORTED MICROSPORES), *Arabidopsis* ortholog of Bra002004, Bra013041, BniB011765, BniB025753, Bol004758 and Bol042692, plays a vital role in tapetum and pollen development (Figure 5) [44]. *Arabidopsis* ortholog *MYB80* (MYB domain protein 80) also called *MS188* (Male Sterile 188), a member of the R2R3 MYB transcription factor gene family, of Bra002847, Bra035604, BniB030464, BniB044215, Bol009875 and Bol035011, is required for tapetal and pollen development (Figure 5) [45]. Bra025337, BniB025201 and Bol042967 are orthologous to *TDF1* (Tapetal Development and Function 1), regulating tapetal differentiation and function (Figure 5) [46]. *Arabidopsis* orthologs *CYP703A2* of Bra032631, Bra033272, BniB042784, Bol018458 and Bol040704, as well as *CYP704B1* of Bra004386, BniB031958 and Bol023932, are essential for sporopollenin synthesis in pollen development (Figure 5) [47,48]. Furthermore, *Arabidopsis* orthologs of some genes (Bra036646 and BniB001185 orthologous to *ACOS5* (Acyl-CoA Synthetase 5), Bra034658, Bra011566, Bra017681, BniB036573, BniB040820, BniB048407, Bol013698 and Bol034656 orthologous to *LAP5* (Less adhesive pollen 5) and Bra017147, BniB016901 and Bol025267 orthologous to *LAP6* (Less adhesive pollen 6)) have been shown to perform important roles in pollen development and sporopollenin biosynthesis (Figure 5) [49,50]. Other important candidate genes Bra010535 and BniB015834 are orthologous to *TKPR1* (TETRAKETIDE alpha-PYRONE REDUCTASE 1) (previously called *DRL1* (Dihydroflavonol 4-reductase-like1)), regulating a metabolic pathway critical for sporopollenin monomer biosynthesis, pollen wall formation and male fertility (Figure 5) [51,52]. Bra004316, Bra008822, BniB031910 and Bol024018 are orthologous to *TKPR2* (TETRAKETIDE alpha-PYRONE REDUCTASE 2), which regulates the biochemical pathway for sporopollenin monomer biosynthesis in *Arabidopsis* (Figure 5) [51]. Bra034793, Bra038691, BniB010680, BniB035126, Bol007277 and Bol010336 are orthologous to *MS2* (Male Sterile 2), which encodes for a plastid-localized fatty acyl carrier protein reductase, is essential for pollen wall development in *Arabidopsis* (Figure 5) [53]. *ABCG26* (ATP-Binding Cassette Transporter G26), *Arabidopsis* ortholog of Bra039378, BniB034970 and Bol015793, is important for male fertility and pollen exine formation (Figure 5) [54]. 

### 2.5. Identification of Preferentially Expressed Genes in Female Gamete Development of Brassica Allohexaploids

We next analyzed the preferentially expressed genes involved in female gamete development to explore their key events in *Brassica* allohexaploids. 4553 genes preferentially expressed in female gamete development were identified by overlapping 7939 and 14,830 upregulated genes of Ovule vs. Anther and Ovule vs. Leaf (Figure 6a). These genes were analyzed by GO biological process pathway analysis. These GO terms related to cell differentiation (GO: 0030154), auxin biosynthetic process (GO: 0009851), integument development (GO: 0080060), cell division (GO: 0051301), glucosinolate biosynthetic process (GO: 0019761), auxin-activated signaling pathway (GO: 0009734), polarity specification of adaxial/abaxial axis (GO: 0009944), regulation of proanthocyanidin biosynthetic process (GO: 2000029), plant ovule development (GO: 0048481) and regulation of jasmonic acid mediated signaling pathway (GO: 2000022) were significantly enriched in female gamete development (Figure 6b). These results suggested that genes preferentially expressed in female gamete development may be related to cell division, nuclear positioning, cell differentiation and auxin-activated synthesis and signal transduction of *Brassica* allohexaploids.

Megagametogenesis begins when FM passes through three rounds of nuclear division, producing an eight-nucleate syncytium. Many genes have been implicated in the mitosis entry and progression during female gametophyte development. *Arabidopsis* ortholog *AUX1* (AUXIN RESISTANT 1) of Bra000160 and Bol020456, and *PIN1* (PIN-FORMED 1) of Bra015983, BniB013785, BniB015990, Bol026255 and Bol039419 are essential in sporophytes for auxin import and local auxin biosynthesis to regulate mitosis (Figure 7) [55,56]. The anaphase-promoting complex/cyclosome (APC/C) is a multi-subunit E3 ligase, which plays a critical role in regulating cell-cycle progression [57]. Bra028755 and BniB038334 are orthologous to *APC1* (Figure 7). The three mutants *apc1-1*, *apc1-2*, and *apc1-3* had similar problems in female gametogenesis, such as degradation, aberrant nuclear number, and altered nuclei polarity in the embryo sac, as well as embryogenesis [57]. *Arabidopsis* ortholog *NOP10* of Bra036475 affects megaspore mitosis and polar nuclear fusion in female gametophyte development (Figure 7) [58]. Bra007730, Bra040432, BniB007136 and Bol045717 are orthologous to *RanGAP1* (Ran GTPase Activating Protein 1), which is essential in sporophytes and may regulate mitotic cell cycle progression of female gametophyte development (Figure 7) [59]. *Arabidopsis* ortholog *TUBG1*, γ-tubulin gene, of Bra007622 is involved in the position of the nuclei (Figure 7) [60]. Bra025912, Bra031027, BniB047288, Bol009730 and Bol030782 are *Arabidopsis* ortholog of *NACK1* (NPK1-ACTIVATING KINESIN 1), which is necessary for cellularization and nuclear positioning during female gametophyte development (Figure 7) [61]. In addition, some genes play an important role in cell specification and development. BniB000251 is orthologous to *NFD1* (Nuclear fusion defective1), encoding the *Arabidopsis* RPL21M protein, which regulates karyogamy in female gametophyte development (Figure 7) [62]. The *Arabidopsis* ortholog *AMP1* (Altered meristem program 1) of Bra014794 and BniB007415 has been shown to play an essential role in synergid cell fate for megagametogenesis (Figure 7) [63].

### 2.6. Validation of Expression Profiling by Real-Time Quantitative PCR (RT-qPCR)

We further tested the reliability of RNA-Seq results by RT-qPCR. The randomly selected male and female gamete development-related genes BniB000031 (*SPO22*), Bra035838 (*RMI1*), Bol022020 (*RECQ4A*), Bra000160 (*AUX1*), Bol039419 (*PIN1*), BniB025753 (*AMS*), Bra025337 (*TDF1*), Bol034656 (*LAP5*) and BniB031910 (*TKPR2*) were verified by RT-qPCR. The results showed that the selected genes were positive by RT-qPCR (Figure 8). The expression patterns showed the same trend as that detected by RNA-Seq analysis (Figure 8). 

## 3. Discussion

### 3.1. Meiosis-Related Genes May Affect Homologous Recombination during Male and Female Gamete Development in Brassica Allohexaploids

The key events in both the male and female gamete development of *Brassica* allohexaploids mainly focused on meiosis-related genes and pathways. Polyploidy species may be used more as model organisms for meiosis in the future, and *Brassica* has attracted attention as model of allopolyploid meiotic regulation mechanisms [64]. Meiosis leads to the production of genetically unique haploid spores, which contribute to genome stability and genetic diversity. Crossover of homologous recombination ensures faithful segregation of homologous chromosomes in meiosis I [65]. Meiotic CO produces new combinations of alleles, increasing the genetic diversity of gametes [66]. Meiosis forms DNA DSB through programming, processes and repairs DSB by homologous recombination. In all eukaryotes, *SPO11-1* is a strict requirement for meiosis DSB formation. In *Arabidopsis* and *Brassica*, meiotic DSB formation is required for synapsis and SC formation. Crossover or non-crossover recombination products can be isolated and genetically tested by recombination intermediates formed between homologous chromosomes. Consequently, we hypothesized that meiosis-related genes may affect homologous recombination during male and female gamete development in *Brassica* allohexaploids.

### 3.2. TDF1, AMS and MS188 May Influence Tapetum Development and Pollen Wall Formation in Brassica Allohexaploids

Male gamete development affects the effective pollination control system, which is the premise of utilizing heterosis of *Brassica* allohexaploids. The development of male gametes is accompanied by tapetal development, sporopollenin synthesis and pollen wall formation. Tapetum is the innermost layer of the four anther somatic cell layers, and provides material for pollen development. The main component of the exine is sporopollenin, which is produced by the tapetum. The tapetal development regulatory network during male gamete development has been studied in *Arabidopsis*, in which *TDF1* may be involved in redox and cell degradation [67,68,69]. *AMS* is thought to control lipid transfer proteins in pollen wall building, repressing of upstream regulators and promoting of AMS protein degradation [44,67]. Most cell wall–related genes are regulated by transcription factor *MS188*, which is involved in both tapetum cell wall degradation and pollen wall synthesis [67]. *TDF1* controls *AMS* directly through an AACCT cis-element, and *TDF1* and *AMS* control downstream genes in a feed-forward loop [68]. In addition, *MS188* can activate the expression of *CYP704B1*, *ACOS5* and *TKPR1*, and form a feedforward loop with its direct upstream regulatory factor *AMS* to activate the sporopollenin biosynthesis pathway and rapidly form pollen wall in *Arabidopsis* [69]. Therefore, we speculated that these genes in *Brassica* allohexaploids may also potentially affect tapetum development and pollen wall formation.

### 3.3. AUX1 and PIN1 May Regulate the Formation of the Seven-Cell Embryo Sac in Brassica Allohexaploids

Female gamete development is rarely studied compared to male gamete development. To enable female gametophyte fertilization, hence plant reproduction, megagametogenesis comprises carefully controlled mitotic divisions, repositioning of nuclei along a polar axis and the acquisition of different identities by individual cells [70]. Auxin levels are known to be controlled by biosynthesis and transport, and it is critical for sporophytic developmental processes. During embryo sac development, localized auxin biosynthesis and import are essential for mitotic divisions, cell growth and patterning [55]. *AUX1* and *PIN1* influence mitosis progression at one-, two- and four-nucleate stages [55]. The final position of the nuclei foreshadows the cellularization pattern that divides the female gametophyte into seven cells. Furthermore, cellular identity is most likely determined by the location of the nuclei and related cells along the micropylar–chalazal axis. Therefore, these genes involved in mitosis, nuclear localization, cell differentiation and development may be related to the formation of mature seven-cell embryo sac in *Brassica* allohexaploids.

## 4. Materials and Methods

### 4.1. Plant Materials, Tissue Collection and RNA Extraction

A trigenomic *Brassica* allohexaploid (BBCCAA, 2n = 54) was generated by interspecific hybridization and chromosome doubling between maternal *B. carinata* (“VI047487”, BBCC, 2n = 34) and paternal *B. rapa* (“JK66-83”, AA, 2n = 20) in this study. These *Brassica* allohexaploid materials were grown in greenhouse with a 16-h light/8-h dark cycle at 22 °C. These plants were later chromosomally identified as allohexaploid (2n = 54). Anthers and ovules of these *Brassica* allohexaploids were collected according to the previous methods reported [25,27]. After flowering, anthers from meiosis to mature pollen stage (Anther) and ovules from meiosis to mature embryo sac stage (Ovule) were collected, and young leaves (Leaf) of main inflorescences were also collected as vegetative tissue (organ) control. Three biological replicates were taken from each tissue of *Brassica* allohexaploids. These samples were rapidly frozen in liquid nitrogen and stored at −80 °C to extract RNA. The purity of the samples was determined by NanoPhotometer^®^ (IMPLEN, Westlake Village, CA, USA). The concentration and integrity of RNA samples were detected using Agilent 2100 RNA nano 6000 detection kit (Agilent Technology, Santa Clara, CA, USA).

### 4.2. cDNA Library Construction, Filter and Alignment

A total amount of 1–3 μg RNA per sample was used as input material for the RNA sample preparations. Sequencing libraries were generated using VAHTS Universal V6 RNA-Seq Library Prep Kit for Illumina ^®^. In order to guarantee the data quality used to analysis, the useful Perl script was used to filter the original data (Raw Data). The reference genomes and the annotation file were downloaded from *B. rapa* genome v1.5 sequence, *B. nigra* genome v1.1 sequence and *B. oleracea* genome v1.0 sequence (http://Brassicadb.cn, accessed on 1 May 2021). Bowtie2 v2.2.3 was used for building the genome index, and Clean Data was then aligned to the reference genomes using HISAT2 v2.1.0 [71]. The RNA-Seq data was uploaded to the NCBI Gene Expression Omnibus (GEO), and its accession number, GSE201456, may be used to retrieve it.

### 4.3. FPKM and DEGs

Reads count for each gene in each sample was counted by HTSeq v0.6.0 (Simon Anders, Heidelberg, Germany), and FPKM was then calculated to estimate the expression level of genes in each sample. DESeq2 estimated the expression level of each gene in per sample by the linear regression, then calculated the *p*-value with Wald test [72]. The *p*-value was corrected by the BH method. Genes with fold change ≥ 2 and q-value (adjusted *p*-value) ≤ 0.05 were identified as DEGs.

### 4.4. Function Enrichment Analysis

The DEGs aligned to COG were classified according to functions of genes (http://www.ncbi.nlm.nih.gov/COG/, accessed on 7 August 2021). The GO enrichment of DEGs was implemented by the hypergeometric test, in which *p*-value was calculated and adjusted as q-value, and data background was genes in the whole genome (http://geneontology.org/, accessed on 27 August 2021). GO terms with q-value < 0.05 were considered to be significantly enriched. GO enrichment analysis could exhibit the biological functions of the DEGs. The information about *Arabidopsis* ortholog of *Brassica* was obtained from Brassicaceae Database (BRAD) (http://Brassicadb.cn, accessed on 1 May 2021). 

### 4.5. RT-qPCR

In order to verify the accuracy of RNA-Seq, nine candidate genes were randomly selected for RT-qPCR. The cDNA used the same RNA samples of the RNA-Seq. The RT-qPCR analysis was performed using the SYBR Green I. To standardize the results, *BrGAPDH* was employed as an internal reference control. Three biological replicates were used for each sample. Primer Premier 5.0 was used to design gene-specific primers for nine genes and these primer sequences were listed in Appendix A. The CT values (the fractional cycle number at which the fluorescence crosses the specified threshold) were generated using CFX manager software and evaluated using the 2^−ΔΔCt^ method [73].

## 5. Conclusions

The transcriptome profiling of anthers and ovules revealed the candidate genes of homologous recombination in male and female meiosis, the genes of tapetum development, sporopollenin biosynthesis and pollen wall formation in male gamete development and the genes of mitosis, nuclear localization, cell differentiation and development in female gamete development of *Brassica* allohexaploids. Our findings enhance the understanding of the genes involved in the male and female gamete development of *Brassica* allohexaploids and lay a foundation for the reproductive study of polyploids.

## Figures and Tables

**Figure 1 plants-11-01556-f001:**
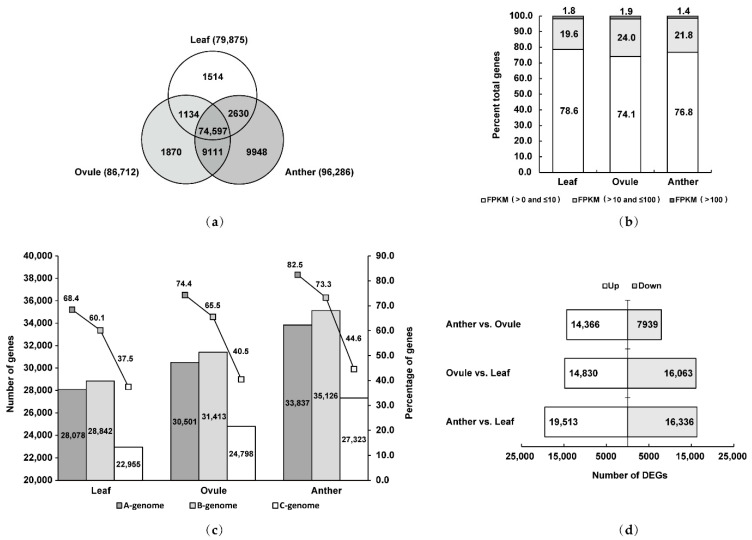
Overview of gene expression in Anther, Ovule and Leaf of *Brassica* allohexaploids: (**a**) Venn diagram showing the overlap between expressed genes in Anther, Ovule and Leaf; (**b**) distribution of FPKM range in Anther, Ovule and Leaf; (**c**) the number of genes from A, B and C-genomes in Anther, Ovule and Leaf and the proportion of genes in the reference genomes; and (**d**) results of pairwise differential expression analysis in Anther, Ovule and Leaf.

**Figure 2 plants-11-01556-f002:**
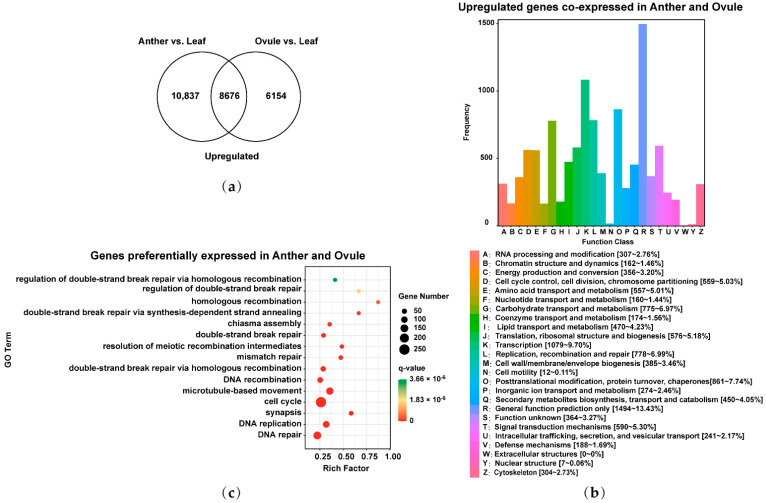
Identification and functional analysis of upregulated genes co-expressed and preferentially expressed genes in Anther and Ovule of *Brassica* allohexaploids: (**a**) Venn diagram showing the overlap of upregulated genes between Anther vs. Leaf and Ovule vs. Leaf; (**b**) clusters of orthologous groups (COG) classification of upregulated genes co-expressed in Anther and Ovule; and (**c**) gene ontology (GO) biological process pathway enrichment analysis of preferentially expressed genes in Anther and Ovule.

**Figure 3 plants-11-01556-f003:**
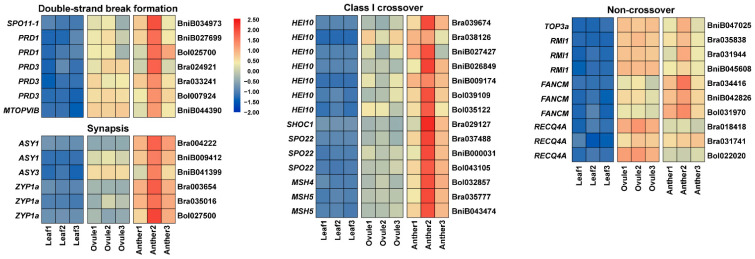
The expression patterns of double-strand break formation, synapsis, class I crossover and non-crossover–related genes in Anther, Ovule and Leaf of *Brassica* allohexaploids.

**Figure 4 plants-11-01556-f004:**
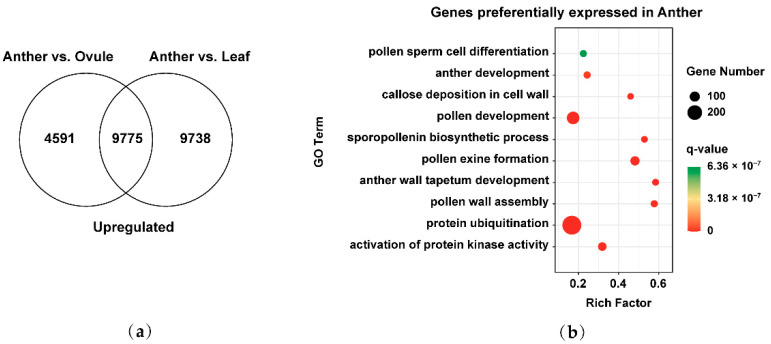
Identification and functional analysis of preferentially expressed genes in male gamete development of *Brassica* allohexaploids: (**a**) Venn diagram showing the overlap of upregulated genes between Anther vs. Ovule and Anther vs. Leaf and (**b**) GO biological process pathway enrichment analysis of preferentially expressed genes in Anther.

**Figure 5 plants-11-01556-f005:**
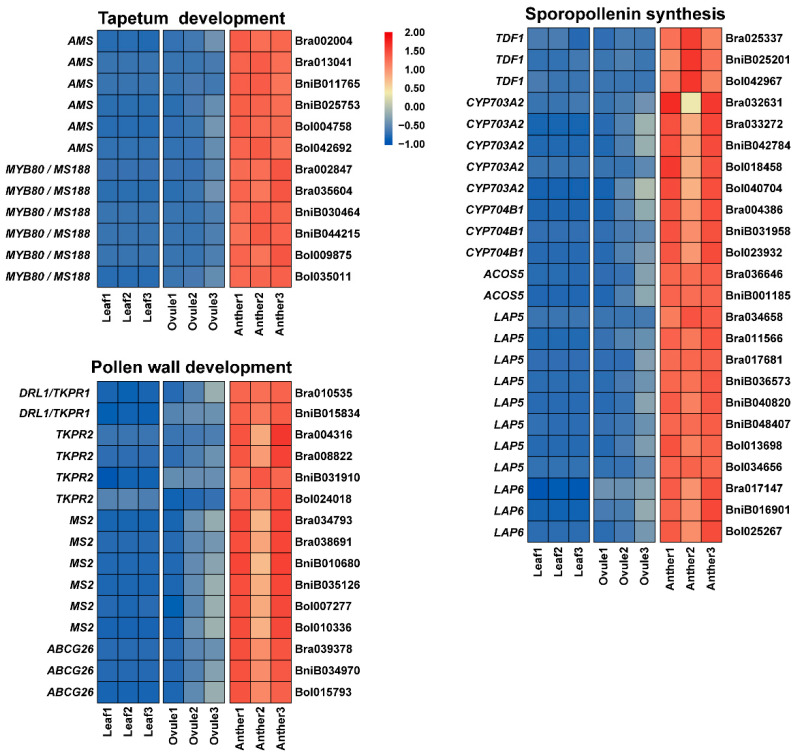
The expression patterns of tapetum development, sporopollenin biosynthesis and pollen wall development related genes in Anther, Ovule and Leaf of *Brassica* allohexaploids.

**Figure 6 plants-11-01556-f006:**
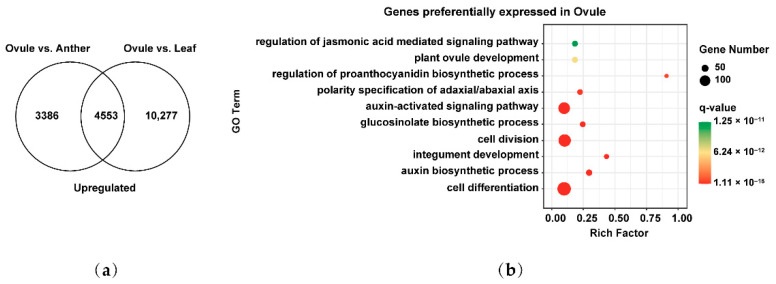
Identification and functional analysis of preferentially expressed genes in female gamete development of *Brassica* allohexaploids: (**a**) Venn diagram showing the overlap of upregulated genes between Ovule vs. Anther and Ovule vs. Leaf; and (**b**) GO biological process pathway enrichment analysis of preferentially expressed genes in Ovule.

**Figure 7 plants-11-01556-f007:**
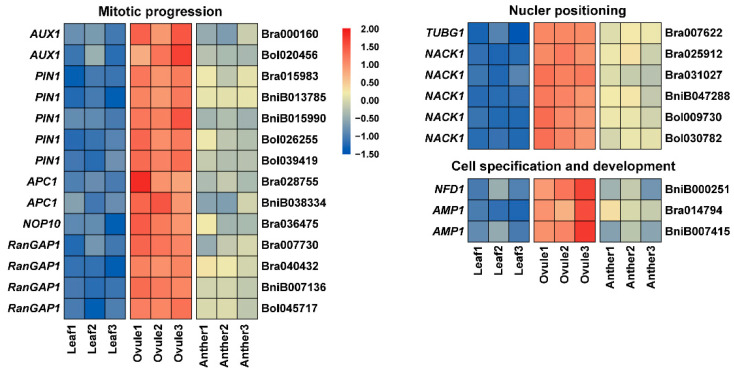
The expression patterns of mitotic progression, nuclear positioning and cell specification and development related genes in Anther, Ovule and Leaf of *Brassica* allohexaploids.

**Figure 8 plants-11-01556-f008:**
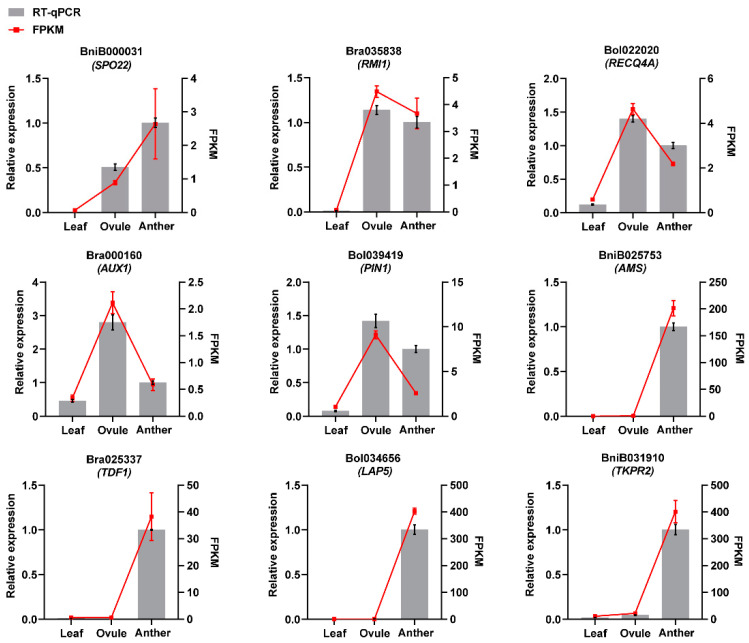
RT-qPCR analysis of nine randomly selected genes. RT-qPCR results are shown by grey columns, and RNA-Seq findings are represented by red lines.

**Table 1 plants-11-01556-t001:** Statistics of the RNA-Seq data in the *Brassica* allohexaploids.

Sample Name	Raw Reads Number	Clean Reads Number	Clean Reads Rate (%)	Raw Q30 Bases Rate (%)	Clean Q30 Bases Rate (%)
Leaf1	49,970,344	46,345,462	92.75	93.65	94.55
Leaf2	45,415,696	40,202,856	88.52	93.22	94.63
Leaf3	47,653,708	44,057,944	92.45	93.56	94.54
Ovule1	51,752,802	45,556,284	88.03	92.76	94.86
Ovule2	49,874,912	46,197,038	92.63	93.56	94.53
Ovule3	48,619,388	45,120,190	92.8	93.36	94.36
Anther1	55,572,362	47,450,126	85.38	91.81	94.29
Anther2	50,943,546	45,328,282	88.98	92.73	94.42
Anther3	50,431,536	46,943,730	93.08	93.67	94.51

**Table 2 plants-11-01556-t002:** Statistics for clean reads mapped in the *Brassica* allohexaploids.

Sample Name	Total Reads Number	Mapped Reads Number	Mapping Rate (%)	MultiMap Reads Number	MultiMap Rate (%)
Leaf1	46,345,462	25,110,254	54.18	991,570	2.14
Leaf2	40,202,856	21,502,559	53.49	868,719	2.16
Leaf3	44,057,944	23,720,073	53.84	978,530	2.22
Ovule1	45,556,284	38,811,219	85.19	1,467,965	3.22
Ovule2	46,197,038	39,142,941	84.73	1,469,346	3.18
Ovule3	45,120,190	38,282,041	84.84	1,488,352	3.30
Anther1	47,450,126	39,735,180	83.74	1,558,104	3.28
Anther2	45,328,282	38,359,513	84.63	1,487,356	3.28
Anther3	46,943,730	39,794,184	84.77	1,508,397	3.21

## Data Availability

The RNA-Seq data were deposited into NCBI GEO and can be accessed with the accession number GSE201456.

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
