# Peer review of "Transcriptome Profiling Identifies Candidate Genes Contributing to Male and Female Gamete Development in Synthetic Brassica Allohexaploids"

_plants, 2022, doi:10.3390/plants11121556_

Round 1
Reviewer 1 Report
Ji et al, have done a lovely paper looking at RNA-seq comparing both female and male gamete development in Brassica allohexploid. This will be a valuable resource for the flowering community as often whole buds are analysed rather than the female and male side separated. It is a shame that only one anther/ovule "stage" was selected, rather than separating out developmental stages but it still gives a lot of insight as by using leaf as a comparison it will highlight all the male or female specific genes throughout their development.
They have performed the analysis and stats expected from an RNA-seq paper, and I only have a few comments that should be addressed;
- I would include the volcano plots from the analysis in the supplemental as well as the Figures shown in Figure 1
- The is no RNA-seq data within the supplemental. While the RNA-seq data has been uploaded into NCBI GEO, I would except at the very least a list of the genes mentioned within the paper (e.g. Fig 3, 5, 7) to have the full RNA-seq data available showing expression data for the 3 tissues/replicates in an excel file. And if possible a full RNA-seq excel file for those that are differentially expressed in anther/ovule - both up and down (e.g. logFC, pValue, FDR for the different genes in the different tissues). So this resource is easy for people to use without having to go back to the original RNA-seq data and doing their own analysis.
- My last comment is without any comparison to genes found in diploid and allotetraploid Brassicas in the same/similar staged material - I do not think that their conclusion reads correctly and should be reworded. How I read their conclusion is that they are suggesting the genes found are important for allohexaploid specifically. Whereas this isnt the case, AMS, DYT1 etc would be also similarly differentially expressed in anther tissue in the diplod and allotetraploid as well. I understand what they are saying and their focus on allohexaploids but I think it comes across in the wrong way in the discussion.
I think it needs to be reworded slightly, or they could open it up to include some RNA-seq studies that have included flowering tissue to show that these genes have also be identified in the diploid/allotetroids B.napus etc - e.g. Liu D., Yu, L., Wei, L., Yu, P., Wang, J., Zhao, H., Zhang, Y., Zhang, S., Yang, Z., Chen, G., et al. (2021). BnTIR: an online transcriptome platform for exploring RNA-seq libraries for oil crop Brassica napus. Plant Biotechnol. J. 19: 1895-1897
Bud2 (for tapetum/meiosis genes)
An analysis of those genes (especially meiosis cross-over/combination etc) in the diploid, allotetraploid and allohexaploid would be very interesting for future work.
Reviewer 2 Report
The manuscript titled “Transcriptome profiling identifies candidate genes contributing to male and female gamete development in synthetic Brassica allohexaploids” is perfectly planned and conducted, and hence should be accepted for publication without minor changes. Please improve the figure quality for 1 and 8. In addition, provide the standard error and deviation for all the figures.
